# Calcite Nanotuned Chitinous Skeletons of Giant *Ianthella basta* Marine Demosponge

**DOI:** 10.3390/ijms222212588

**Published:** 2021-11-22

**Authors:** Ahmet Kertmen, Iaroslav Petrenko, Christian Schimpf, David Rafaja, Olga Petrova, Viktor Sivkov, Sergey Nekipelov, Andriy Fursov, Allison L. Stelling, Korbinian Heimler, Anika Rogoll, Carla Vogt, Hermann Ehrlich

**Affiliations:** 1Center of Advanced Technology, Adam Mickiewicz University, Uniwersytetu Poznańskiego 10, 61-614 Poznan, Poland; ahmker@amu.edu.pl (A.K.); iaroslavpetrenko@gmail.com (I.P.); 2Institute of Electronics and Sensor Materials, TU Bergakademie Freiberg, Gustav-Zeuner-Str. 3, Raum 307, 09599 Freiberg, Germany; andriyfur@gmail.com; 3Institute of Materials Science, TU Bergakademie Freiberg, Gustav-Zeuner Str. 5, 09599 Freiberg, Germany; schimpf@iww.tu-freiberg.de (C.S.); rafaja@ww.tu-freiberg.de (D.R.); 4Institute of Physics and Mathematics of Federal Research Centre Komi Science Center Ural Division of the Russian Academy of Sciences (IPM FRC Komi SC UrB RAS), 167982 Syktyvkar, Russia; teiou@mail.ru (O.P.); sivkovvn@mail.ru (V.S.); nekipelovsv@mail.ru (S.N.); 5Department of Chemistry and Biochemistry, The University of Texas at Dallas, 800 W Campbell Rd, Richardson, TX 75080, USA; stelling@utdallas.edu; 6Institute of Analytical Chemistry, TU Bergakademie Freiberg, 09599 Freiberg, Germany; korbinian.heimler@chemie.tu-freiberg.de (K.H.); anika.rogoll@chemie.tu-freiberg.de (A.R.); carla.vogt@chemie.tu-freiberg.de (C.V.)

**Keywords:** biological materials, biomineralization, calcite, chitin, sponges, scaffolds

## Abstract

Marine sponges were among the first multicellular organisms on our planet and have survived to this day thanks to their unique mechanisms of chemical defense and the specific design of their skeletons, which have been optimized over millions of years of evolution to effectively inhabit the aquatic environment. In this work, we carried out studies to elucidate the nature and nanostructural organization of three-dimensional skeletal microfibers of the giant marine demosponge *Ianthella basta*, the body of which is a micro-reticular, durable structure that determines the ideal filtration function of this organism. For the first time, using the battery of analytical tools including three-dimensional micro—X-ray Fluorescence (3D-µXRF), X-ray diffraction (XRD), infra-red (FTIR), Raman and Near Edge X-ray Fine Structure (NEXAFS) spectroscopy, we have shown that biomineral calcite is responsible for nano-tuning the skeletal fibers of this sponge species. This is the first report on the presence of a calcitic mineral phase in representatives of verongiid sponges which belong to the class Demospongiae. Our experimental data suggest a possible role for structural amino polysaccharide chitin as a template for calcification. Our study suggests further experiments to elucidate both the origin of calcium carbonate inside the skeleton of this sponge and the mechanisms of biomineralization in the surface layers of chitin microfibers saturated with bromotyrosines, which have effective antimicrobial properties and are responsible for the chemical defense of this organism. The discovery of the calcified phase in the chitinous template of *I. basta* skeleton is expected to broaden the knowledge in biomineralization science where the calcium carbonate is regarded as a valuable material for applications in biomedicine, environmental science, and even in civil engineering.

## 1. Introduction

Biominerals in unicellular and multicellular organisms serve many functions in both their survival and active metabolic existence. It is not surprising that in a multi-million years of evolutionary process, organisms have developed the ability to use a wide range of different chemical elements to create biominerals. Thus, a variety of organisms sequester not only Ca and Si, but also Au, Zn, Mn, Cr, Ni, V, Fe, into biominerals (see for an overview [1]).

Biomineralization in sponges (Porifera), which were the first multicellular organisms on the planet, remains a focus of modern biomineralogy and bioinspired materials chemistry. Such poriferan classes as Hexactinellida, Demospongia, and Hexascleromorpha include species with silicified skeletal constructs (spicules and frameworks). In some cases, representatives of hexactinellids, or glass sponges, possess nanophases of calcite within their siliceous skeletal structures [2]. Unique silica-aragonite-chitin biocomposites are examples of multiphase biomineralization that have been discovered in selected verongiid demosponges [3]. However, sponges that belong to the Calcarea class are known to construct their hard tissues only from calcium carbonates [4,5]. While evidence for the presence of the aforementioned biominerals in sponges′ skeletons is numerous and indisputable, debate still revolves around understanding the principles of biomineralization and corresponding mechanisms at the molecular level. Particular attention is paid to the identification of proteins (i.e., silicateins, glassin, cathepsins, collagens) [6,7,8,9] and polysaccharides (i.e., chitin) [10,11] that act as templates for the formation of silica-based mineral phases. The forehead sponge body is a soft and elastic organic matter resting on the mineral skeleton of a particular physical form. This functional biological 3D construct is designed to extract appropriate feed from the aquatic environment by filtration using micro- and macropores with maximum efficiency. Accordingly, the mineralization of the sponge skeleton contributes to an increase in the total area of the filtering surface and the adoption of an optimal position for the sponge body in water, being attached to a solid substrate.

In this study, we have focused our attention on one of the largest fan-shaped marine demosponges *Ianthella basta* (Pallas, 1766) that can reach up to 2.5 m both in height and in its diameter [12] (Figure 1). Previously, ianthellids have been intensively studied due to the biosynthesis of diverse biologically active bromotyrosines [13,14,15,16] as well as their unique flat 3D network-like chitin-based skeleton [17,18]. This kind of naturally pre-structured chitin has been reported as a ready-to-use renewable biomaterial for a broad variety of applications in tissue engineering [19,20,21,22,23], extreme biomimetics [24,25,26], and biomedicine [18,27,28]. In contrast to other representatives of the Verongiida order where corresponding biomineral phases have been already reported [3,29], there are still no reports on biominerals within skeletal structures of ianthellids, including *I. basta* demosponge. At the same time, the ability of this sponge to maintain a vertical position in water space, being firmly attached to the substrate (Figure 1), logically suggests the presence of some kind of “cementing” component inside this structure. Also, previously we have observed the development of microbubbles after insertion of this sponge into acidic solutions that suggest the occurrence of some calcium carbonates. Thus, this study aims to carry out identification of the suggested biomineral phase, which is localized in the skeletal micro-fibers of this demosponge, and to propose a possible biomineralogical scenario for its appearance.

## 2. Results

### 2.1. Identification of Ca and Br Localization Using Three-Dimensional Confocal Micro X-ray Fluorescence (CµXRF)

Recently, the CµXRF has been successfully used in studies on the chemistry of spongin-based skeletal fibers of marine demosponge *Hippospongia communis* [30]. Consequently, we have used the same method for the investigation of the spatial distribution of selected elements within chitinous skeletal structures of *I. basta* demosponge in three samples: as collected after washing with distilled water, and treated with HCl and with NaOH, respectively. We took the liberty to obtain information about the distribution of Br (as the main element in skeletal structures of verongiids [31]) and Ca; though, the elements that we have previously suggested are also present.

In 2D-µXRF the intensity of the elemental signals is heavily influenced by the sample thickness, thus misinterpretation of elemental distribution in structurally inhomogeneous samples may occur. Therefore, elemental distribution was re-analyzed by 3D analysis (after 2D-analysis) in order to define a three-dimensional probing volume, which is formed by the confocal arrangement of the spectrometer’s optics. With this visualization of the elemental distribution of calcium and bromine and a determination of any differences in their spatial distribution was possible. The 3D mapping allowed us to determine the dependence of the elemental distribution up to an absolute analysis depth of 1500 µm, as well as the impact of treatment with different chemicals (Figure 2).

3D visualization of the samples under study showed that the maximum Ca-Kα signal intensities of the original *I. basta* skeleton fragment treated with water and that after alkali treatment was 40 and 19 cps (counts per second), respectively. After the treatment of the same specimen with 3M HCl, the calcium in the skeletal scaffold was dissolved away and nearly no calcium was observed, resulting in a drastically reduced signal intensity of 2 cps, which is more than an order of magnitude lower compared with the calcium signal values of the two samples listed above. However, it seems that the different solution treatments have no significant effect on the absolute Br-Kα signal intensity with 38.0 cps (Figure 2a), 34.0 cps (Figure 2b), and 32.0 cps (Figure 2c), indicating, that each element is bound in different ways in the skeletal scaffolds.

It is noteworthy to highlight that thanks to the presence of Br in all the skeletal samples under study and the high fluorescence energy of the Br-Kα fluorescence lines, it was possible to obtain signals from 1500 µm depth with the applied method. Thus, the 3D visualization obtained from Br analysis does not only represent the surface morphologies but also the in-depth architecture of the sponge scaffolds and therefore, the structural complexity of the *I. basta* skeletal scaffolds was revealed to a greater extent. However, in comparison to bromine, Ca is a much lighter element and therefore the energy of the Ca-Kα fluorescence lines is significantly lower. This results in a bigger cross-section dimension of the probing volume of Ca (28 µm for Ca-Kα, 16 µm for Br-Kα) and a much more limited measurement depth due to a more intensive absorption of the Ca fluorescence radiation by the sponge matrix. Consequently, the actual *I. basta* skeletal structure seems to be encircled by a Ca-containing “shell”. This suggestion has been confirmed by observations of the same structures using SEM (Figure 3a–d).

### 2.2. Visualization of the Presence of Mineral Phase Using Electron Microscopy Methods

To obtain knowledge about the micro-and nano-architecture of the original, water-rinsed fragments of the *I. basta* skeleton (see Figure 2a) we studied them initially with SEM. It was possible to visualize the existence of water-insoluble nanoparticles, which tightly cover the surface of the skeletal fibers. Further EDS analysis of these fibers confirmed the presence of Ca together with other elements typical for skeletal structures of verongiid sponges [2] (Figure 3a).

Investigations of isolated nanoparticles using TEM strongly confirmed the localization of Ca within them (Figure 4b,c). Furthermore, additional TEM studies showed that these nanoparticles are nanocrystalline (Figure 5) that opens the way for corresponding X-ray diffraction analysis to identify this mineral phase.

### 2.3. Identification of Calcite within I. basta Skeletal Fibres

To identify the nature of the mineral phase under study, we have used classical analytical methods such as X-ray diffraction (XRD), Fourier transforms infrared (FTIR), and Raman spectroscopy. These methods have been recognized as absolutely appropriate for the identification of diverse calcium carbonates of biomineral origin [2,3,32,33]. XRD and subsequent qualitative phase analysis revealed the presence of calcite (CaCO_3_, ICSD PDF# 00-005-0586, space group *R*$\bar{3}$*c*) in the samples under study. The diffraction pattern in Figure 6 shows the measured pattern (dots) and the refinement result as solid lines. The refinement was done as a standard Rietveld refinement but without the constraint for the relative intensities, known as the Pawley fit. The non-matching relative intensities result most likely from an insufficient powder average (bad grain statistics). However, the contribution of the calcite to the overall diffraction pattern is highlighted in Figure 6. The lattice parameters of calcite were determined as a = (4.976 ± 0.002) Å and c = (16.981 ± 0.009) Å. The experimentally determined lattice parameters are thus slightly smaller than those from the PDF reference cited above (a = 4.989 Å, c = 17.062 Å).

Along with calcite, quartz (SiO_2_, ICSD PDF# 00-046-1045, space group *P*3_2_21) was found in the sample and a series of minor unidentified peaks, which belong most likely a couple of organic phases which we were unable to identify from our PDF4 database search.

Additionally, we have identified calcite within discovered nanocrystalline mineral phase using infra-red and Raman spectroscopy that is a well-recognized method for such analysis. Correspondingly, the band at 712–713 cm^−1^, which was well visible in the investigated specimens which are characteristic for calcite standard (Figure 7), was found in agreement with the data reported previously for biogenic calcite [34].

The data obtained using FTIR spectroscopy also correlated well with results from Raman spectroscopy investigations (Figure 8) of the same samples. Characterization of the calcium carbonate phase isolated from the skeletal fibers of *I. basta* demosponge with Raman spectroscopy was also in agreement with the data reported in the literature previously [35,36].

### 2.4. Near Edge X-ray Fine Structure (NEXAFS) Features of Identified Mineral Phase

In our work, we used the building block picture for the ionic crystal of CaCO_3_ in order to gain insight into the nature of the structural components. As shown in [37], a preliminary analysis of complex structures can be obtained within a building-block model, by which the structure is seen as an assembly of smaller pieces. Those building blocks contribute independently to the total spectrum. In this way, specific spectral features can be correlated with contributions arising from individual functional groups. The convincing evidence that skeletal fibers of *I. basta* demosponge contains calcium carbonate as a mineral was obtained for the first time from NEXAFS C1s and Ca2p experiments. The NEXAFS in the regions of the C1s and the Ca2p edges are diagnostic for the identification of the calcium carbonate mineral.

In the mineral calcite, and its polymorphs–vaterite and aragonite, there is an ionic bond between the calcium cation and the planar anion [CO_3_]^2−^. Inside the anion, carbon and oxygen atoms are in a covalent bond. Thus, a stable flat group [CO_3_]^2−^ is a necessary element of calcite. Moreover, in calcite, calcium is in the immediate environment of 6 oxygen atoms which are close to octahedral. In vaterite, the calcium environment is also consisting of 6 atoms and is similar to octahedral, which determines the strong similarity of the fine structure of Ca 2p spectra of these minerals [38].

In aragonite, the calcium environment is completely different and amounts to 9 oxygen atoms, and, accordingly, the fine structure of its Ca 2p spectrum is different from vaterite and calcite [38]. It is important to note that in calcium oxide CaO, the environment of calcium with oxygen is the undistorted octahedron and the fine structure of the Ca 2p spectrum of CaO is the same as of the calcite. The Ca 2p spectra themselves reflect only the symmetry of the nearest oxygen environment; therefore, the [CO_3_]^2−^ anion availability is evidence for the presence of calcite and its polymorphs. The narrow π * resonance (290.1–290.3 eV) in the NEXAFS C1s absorption spectrum demonstrates the presence of the planar group [CO_3_]^2−^ [39,40].

The NEXAFS in the regions of the C1s and the Ca2p edges are used for the identification of the calcium carbonate mineral (Figure 9). In calcite, the carbon atoms are in a planar CO_3_ group, and the Ca atoms are in an octahedral coordination environment of the oxygen atoms. The presence of a carbonate peak at 290.28 eV in the C K-edge NEXAFS spectrum of the *I. basta* before and after treatment (Figure 9 left) indicates that this peak is closely associated with carbonates (290.2 eV) [39].

In Figure 9 left the presence of this peak in the spectra of calcite and the sponge processed in alkali is visible, which undoubtedly shows the presence of calcite and possibly vaterite in its composition. Moreover, this peak is absent in the C1s spectrum of the initial *I. basta*, which is due to the low concentration of calcites on its surface and the significantly lower probability of C1s–C2p transitions in the [CO_3_]^2−^ anion as compared to 2p–3d transitions in the calcium atom. The NEXAFS C1s spectrum of the initial sponge corresponds to chitin [41,42], a similar structure is found in the spectrum of the sponge processed in alkali since it contains residual chitin. As for the structure in the region of 284–288 eV in the C1s spectrum of the calcite standard, its presence is due to carbon-containing impurities introduced into the sample upon rubbing the calcite bulk in the metal holder of the sample in air.

The NEXAFS Ca2p spectra, with all four electron transitions, match properly with the calcite spectrum. At that, the Ca 2p-edge NEXAFS peak positions obtained for cleaned *I. basta* skeletal fibers are located at 347.9, 349.3, 351.3, and 352.5 eV (Figure 9 right). All of them exactly correlate with the data obtained from the calcite standard and of those reported for NEXAFS spectra of calcite in the literature [38]. It can be seen (Figure 9 right) that the Ca2p TEY signal intensity of the distilled water washed *I. basta* sample is very small, due to the small concentration of calcium atoms on its surface, approximately 15 times smaller than the calcite standard. After treating the sponge in an alkali solution, the calcite concentration in the sample increases, and the TEY signal intensity increases sharply. The NEXAFS Ca2p spectrum becomes stronger and completely coincides with the NEXAFS spectrum of the calcite standard.

## 3. Discussion and Conclusions

In this study, we showed for the first time that biomineral calcite in the form of a nanoparticulated phase is located in the surface layer of marine demosponge *I. basta* skeletal fibers, which are made of chitin [17,18]. It is noteworthy that the sponge specimens have been collected from two different geographical areas (Guam, USA, and Australia). Experimental data represented here confirm the complex chemistry of skeletal structures of ianthellids which contain mostly halogens (Br, I, Cl) [15]; as well as Na and Ca (Figure 3a). The presence of halogenated compounds in these demosponges has been suggested to play a protective role against diverse microorganisms [15,31]. Recently, it was shown that such brominated compounds as bromotyrosines are localized within so-called spherulocytes, highly specialized cells which are responsible for the biosynthesis of these very effective antivirals [43], antibacterial [31], and cytotoxic [44,45] compounds. These cells have been reported within the outermost layers of corresponding skeletal fibers. While the localization of Br and Ca in the same layers has been shown by us using CµXRF, we cannot yet speculate about the possible role of brominated compounds in calcification. In contrast, the templating activity of chitin with respect to the formation of calcium carbonate-based biominerals is well recognized [3,46,47,48]. Based on chitin identification within the calcitic phase under study using the highly sensitive NEXAFS method (Figure 9), we hypothesize that this structural amino polysaccharide plays a crucial role in calcification within skeletal constructs of *I. basta* demosponge. A schematic overview is represented in Figure 10.

The origin of calcium that is necessary for the formation of calcite in our case is unclear. It could be from surrounding seawater, or the rocky substrate as this sponge is able to dissolve it during ontogenesis using an enzymatic reaction (carbonic anhydrases-based) similar to that of calcite producing adhesive diatoms [49]. The presence of calcite in chitinous fibers seems to be reasonable: incorporation of this nanocrystalline biomineral phase into *I. basta* demosponge skeleton appears to be selectively favored because the resulting biological material becomes much more rigid than mineral-free skeletons of spongin-producing demosponges. Consequently, this large-sized sponge must possess a highly specific porous skeleton with mechanical properties that can provide both support for a large internal surface area and stability underflow currents of the open sea similar to giant tube-like skeletons of *Aplysina archeri* verongiid demosponge as reported by us previously [50]. Thus, calcification plays a crucial role in the survival of the whole sponge. However, the occurrence of calcium carbonate in skeletal structures of *I. basta* under conditions of ocean acidification can be dangerous due to naturally occurring demineralization. One of the most intriguing questions that shall be addressed by further investigations is to decipher the natural selection with respect to calcite (as in calcarean sponges) in the case of *I. basta* demosponge and not to aragonite, or vaterite. Additionally, the origins of the sphere-like morphology observed in the nanostructured calcite that is responsible for biomineralogical nano-ornamentation of sponge skeleton remains unclear.

Understanding the biomineralization process has long attracted the attention of researchers for its potential implications in designing novel composite materials and establishing new synthetic methods for mimicking the naturally occurring properties of the biominerals [51,52]. It has been long-discussed that the development of bioinspired functional inorganic-organic hybrid materials with ultralightweight, high flexibility and mechanical strength could be only facilitated by tuning the crystal morphology, size, composition, and hierarchical organization of materials [53]. Since it has been clearly outlined that such advanced materials could be more likely achieved by a multidisciplinary approach and collaborative research [53], the findings presented herein are expected to contribute to the development of the field from the biological, physiochemical, and material properties point of view. We believe our research findings along with a hypothetical schema we presented for the chitin-templated formation of calcite in *I. basta* could illuminate a broader understanding of the biomineralization process. Calcite, as one of the most widespread biominerals of nature, already has numerous known applications in biomedicine [54,55,56], civil engineering, and environmental sciences [57]. Relatedly, the discovery of the chitin-templated calcification process in *I. basta* demosponge as a renewable source of unique network-like scaffolds could bring another dimension to the applicability and reproducibility of calcium-based biomaterials in similar applications.

## 4. Materials and Methods

### 4.1. Supply of Demosponge Specimens

Samples of *Ianthella basta* (Pallas, 1766) (Demospongiae, Verongiida, Ianthellidae) marine demosponge have been collected underwater from depths of 5–12 m by scuba diving in two different regions: at Western Shoals in Apra Harbor (Guam) in fall 2008 and spring 2009 (see for details [17] as well at station SOL47/W/A042 (1536′46.10″ S, 12404′22.92″ E to 1536′44.77″ S, 12404′22.38″ E, Kimberley, Western Australia in March 2015 at a depth of 35.3–35.5 m.

### 4.2. Preparation of Demsoponge Samples for Analysis

Selected *I. basta* sponge samples were prepared by different solution-based treatment methods. The first group of samples was solely stored in distilled water for 28 h at room temperature to remove water-soluble salts and compounds. The second group of sponges have been treated with 3M HCl for 28 h at 24 °C to remove possible calcium carbonates and acid-soluble organic matter. The HCl solution was refreshed 2-times after 5 h and 16 h of treatment. 7 h after the last change of the HCl solution, the sample was washed three times with distilled water. The third group of *I. basta* skeletons were treated with 10% NaOH at 40 °C for 28 h and finally washed three times with distilled water. All the samples were dried in an oven for 16 h at 40 °C.

### 4.3. Digital, Light, and Fluorescence Microscopy

The samples were observed and analyzed with the use of an advanced imaging and measurement system consisting of a Keyence VHX-6000 digital optical microscope and VH-Z20R swing-head zoom lenses (magnification up to 200×).

### 4.4. FTIR

Infrared spectra were recorded with a Perkin Elmer FTIR Spectrometer Spectrum 2000, equipped with an AutoImage Microscope using the FT-IRRAS technique (Fourier Transform Infrared Reflection Absorption Spectroscopy).

### 4.5. Raman

FT-Raman spectra were measured using a Bruker RFS 100/s spectrometer and Nd-YAG excitation at 1064 nm.

### 4.6. Scanning Electron Microscopy (SEM) and Energy-Dispersive X-ray Spectroscopy (EDS)

The samples under study were fixed in a sample holder and covered with carbon, or with a gold layer for 1 min using an Edwards S150B sputter coater (BOC Edwards, Wilmington, MA, USA). The ready-to-use specimens were then placed in an ESEM XL 30 Philips or LEO DSM 982 Gemini (Cambridge, UK) scanning electron microscope. The energy-dispersive X-ray spectroscopy (EDS) was carried out at 20 kV on a JEOL scanning electron microscope JSM-6610LV (LaB6 filament) at a working distance of 10.0 mm with Bruker XFlash 6|10 silicon drift detector (SDD, detector area of 10 mm^2^) with an energy resolution of 123.8 eV at Mn-Kalpha and the detector window AP3.3.

### 4.7. Transmission Electron Microscopy (TEM)

The microstructure was determined using the Hitachi S4700 and XL 30 ESEM (Fa. Philips Electron Optics GmbH, Hamburg, Germany) scanning electron microscope. Samples were dried and covered with Au using a Cressington 108 Auto sputter coater.

### 4.8. Near Edge X-ray Fine Structure (NEXAFS)

The electronic structure of selected skeleton fragments of *I. basta* marine demosponge after the cleaning procedures described above were characterized by the means of the near-edge X-ray absorption fine structure (NEXAFS) spectroscopy at the Berliner Elektronenspeicherring fur Synchrotronstrahlung (BESSY) using radiation from the Russian–German dipole beamline [58]. The NEXAFS C1s и Ca2p spectral data provide information about p- and d- like the partial density of unoccupied electronic states according to the dipole selection rules. The corresponding electronic states are localized on the atoms, where the excited electrons were initially bound. The NEXAFS spectra are element-specific because each atom has characteristic binding energies of core levels. The unoccupied, element-specific p- and d- character electronic states of the selected samples and the calcite standard were probed by taking the C1s and Ca2p absorption spectra. All near-edge X-ray absorption fine structures were acquired in the total electron yield (TEY) mode. The energy calibrations were made by using the well-resolved resonance at 285.38 eV of the C1s spectrum of HOPG [59] for the NEXAFS C1s-spectrum of the sample. At that, the energy resolution was less than 0.1 eV. The signal of the sample was normalized to the incident radiation intensity determined by the Au plate TEY signal according to a previously described procedure [60].

### 4.9. XRD Analysis

X-ray diffraction was performed with the purpose of phase identification. The X-ray diffraction pattern was recorded with a Seifert/FPM RD7 diffractometer equipped with a sealed X-ray tube with Cu anode. The experiment was performed in symmetrical Bragg-Brentano diffraction geometry. The powder sample was put on a zero-background holder (Si, <510> cut). The diffracted beam passed a set of slits and a graphite monochromator before being detected by a proportional counter.

### 4.10. Confocal Micro X-ray Flourescence (CµXRF)

The CµXRF analysis was performed on a modified M4 TORNADO µXRF spectrometer (Bruker Nano GmbH, Berlin, Germany). The spectrometer was equipped with a 30 W Rh-Tube (metal-ceramic micro-focus tube), operated at a high voltage of 50 kV and an anode current of 600 µA. The commercial µXRF setup was equipped with a polycapillary lens with a spot size ≤20 µm (Mo-Kα) for X-ray focusing and a 30 mm^2^ silicon drift detector (SDD). Due to a modification, a second polycapillary lens was installed perpendicular to the first one in front of a 60 mm^2^ SDD. For three-dimensional analysis special sample preparation in the course of the complex naturally grown sponge structure needed to be applied. The sponge samples were fixated on the bottom of a plastic frame, which was filled with small glass marbles, for the stabilization of the sample. The samples were analyzed by measuring 101 XY-mappings inside a sample volume of 1.5 × 1.5 × 1.5 mm^3^. Each mapping was measured with a spot distance of 15 µm and a spot measuring time of 30 ms. Given the integrity of the sample, no vacuum was applied to the sample chamber. The produced two-dimensional datasets consisting of three dimensions, location coordinates X, Y, and the signal intensity, were normalized and converted into RGB color-coded images via in-house software. The stacking of the individual elemental distribution images was carried out with the image editing program ImageJ. With the plugin Volume Viewer, the two-dimensional information of the stackings was converted into a three-dimensional image. The Volume Viewer plugin was used with an image sampling number of 5.0, a tricubric smooth interpolation for visualization, and the RGB color space. µXRF analysis was performed by point measurements with measuring time of 120 s, 50 kV acceleration voltage, and 200 µA anode current at atmospheric pressure.

## Figures and Tables

**Figure 1 ijms-22-12588-f001:**
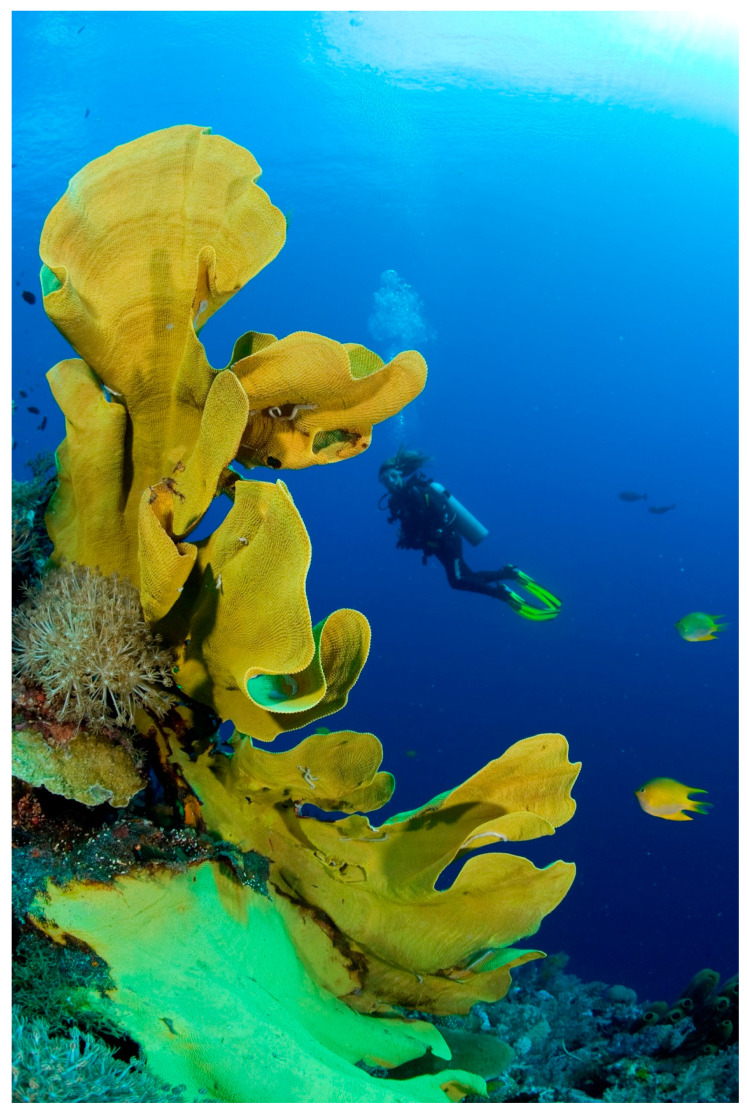
Marine verongiid demosponge *Ianthella basta* possess a huge chitin-based, a network-like skeleton that is stiff enough to withstand underwater currents due to its natural rigidity.

**Figure 2 ijms-22-12588-f002:**
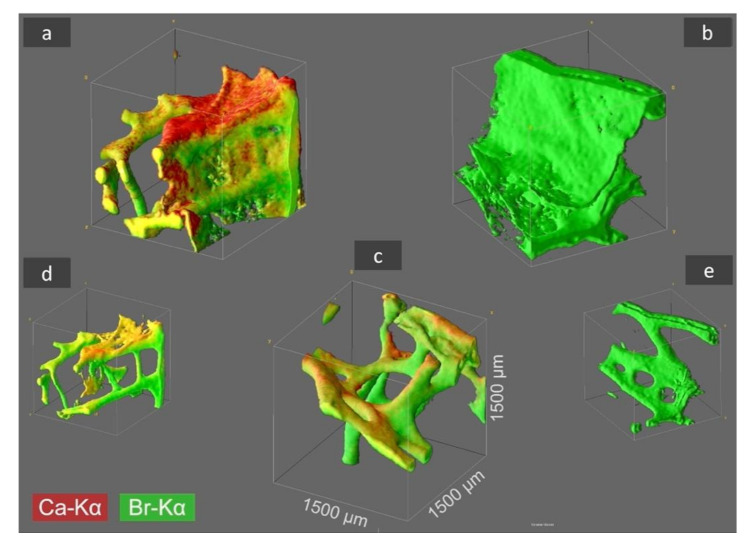
CµXRF analysis results of selected *I. basta* skeletal samples after treatment with distilled water (**a**), 3 M HCl (**b**), and 10% NaOH at 40 °C (**c**). The 3D distribution of Ca (red) and Br (green) is illustrated in the figure for an analyzed sample volume of 1.5 × 1.5 × 1.5 mm^3^ (and locations with both elements are shown in a yellow coloration). By HCl treatment the Ca is removed from the skeletal structure (**b**), whereas by NaOH treatment both elements, Ca and Br, remain within the demosponge, however thin proteinaceous tissue layers between the sponge’s skeleton are dissolved (**c**) and characteristic 3D chitinous construct remains present [17]. In figures (**d**,**e**) the sponge structure of (**a**,**b**) is depicted without the above-mentioned thin tissue layers, which was obtained by omitting the regions with very low signal intensity from data processing.

**Figure 3 ijms-22-12588-f003:**
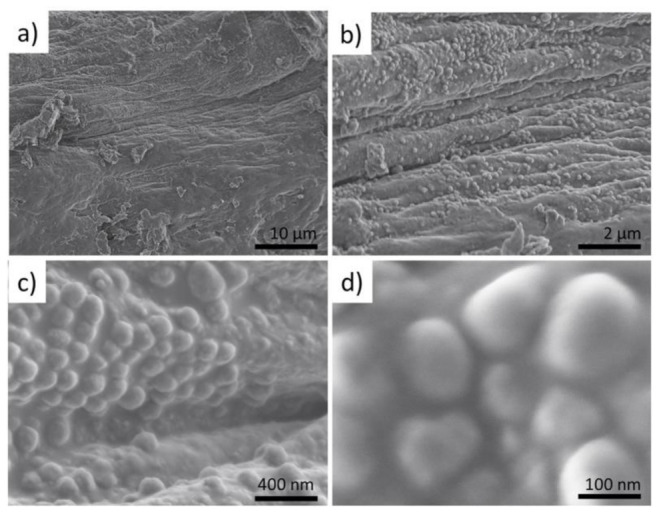
SEM images of the selected fragment treated with a distilled fragment of *I. basta* sponge skeletal fiber (**a**) show the presence of sphere-like nanoparticles (**b**–**d**).

**Figure 4 ijms-22-12588-f004:**
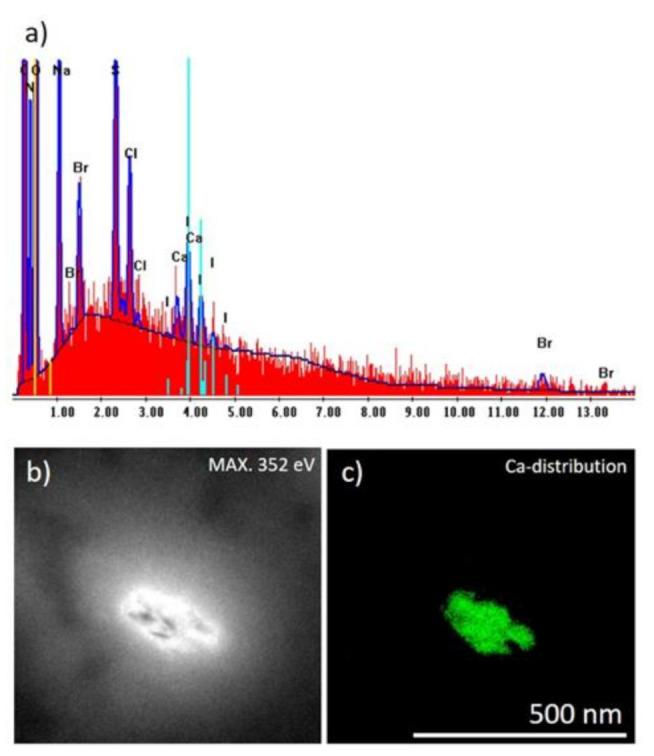
(**a**) EDS analysis of the *I. basta* skeletal fiber washed with distilled water demonstrating the atomic composition of this structure; (**b**) bright-field image of corresponding TEM image and (**c**) Ca distribution within the corresponding sample visualized in green color (colorization provided by the software to highlight electron diffraction signals of Ca atoms).

**Figure 5 ijms-22-12588-f005:**
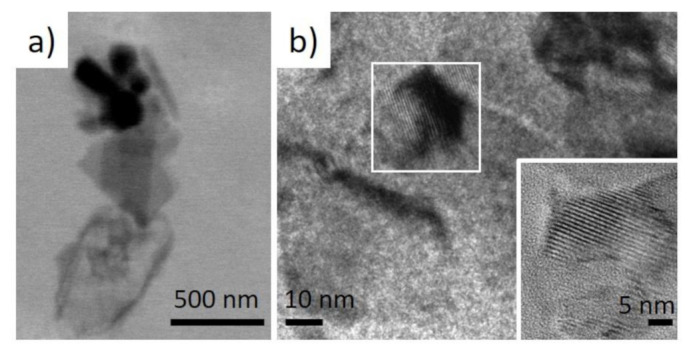
(**a**) low-resolution TEM image of the selected micro-fragment of the *I. basta* skeletal fiber-containing nanoparticles and (**b**) high-resolution TEM images showing their crystalline nature.

**Figure 6 ijms-22-12588-f006:**
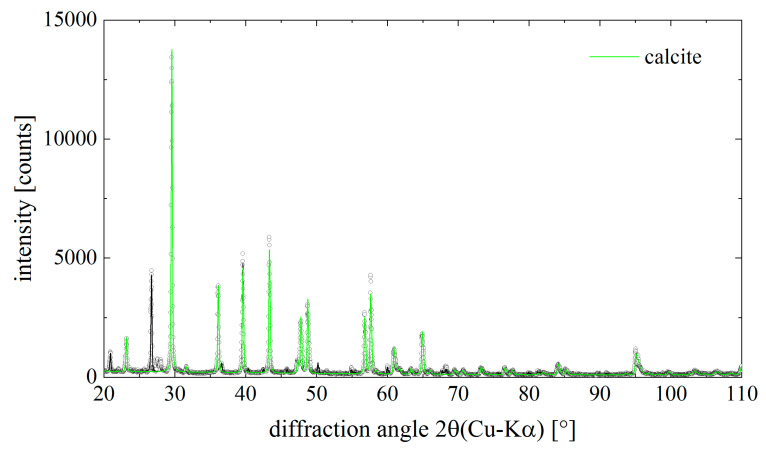
Diffraction pattern of the mineral phase isolated from chitinous skeletal fibers of *I. basta* demosponge. Dots are measured data points, the solid lines result from a Rietveld- (Pawley)-fit of the data. The contribution of calcite to the total diffraction pattern is highlighted in green color. Calcite is the main component in the sample.

**Figure 7 ijms-22-12588-f007:**
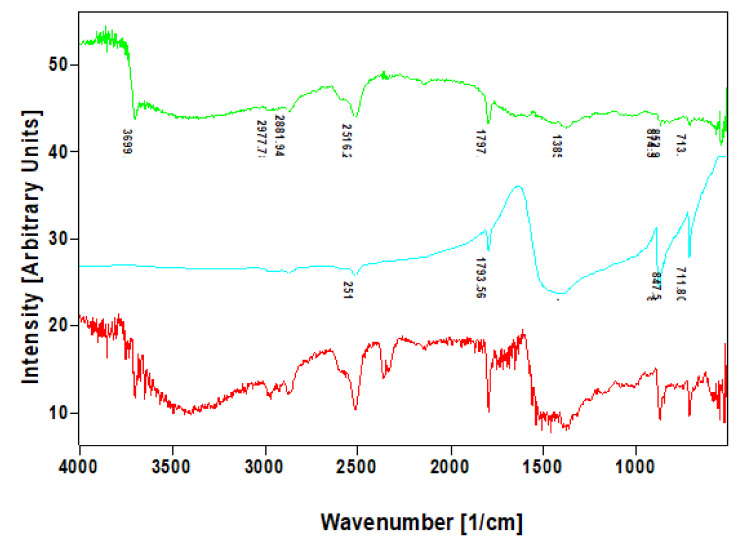
Comparative FTIR spectra of calcite standard (blue line) with that of nanocrystalline phases isolated from skeletal fibers of *I. basta* demosponges collected near Guam (USA) (red line) and in coastal waters of Australia (green line).

**Figure 8 ijms-22-12588-f008:**
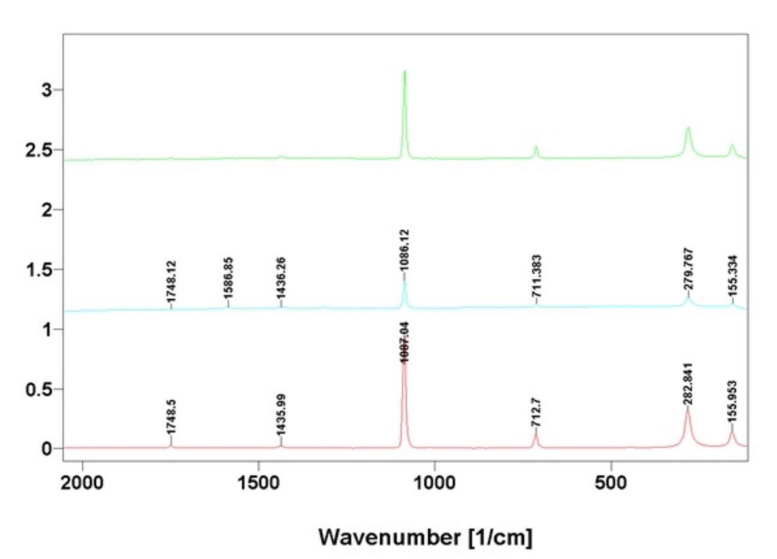
Comparative Raman spectra of calcite standard (blue line) with that of nanocrystalline phases isolated from skeletal fibers of *I. basta* demosponges collected near Guam (USA) (red line) and in coastal waters of Australia (green line).

**Figure 9 ijms-22-12588-f009:**
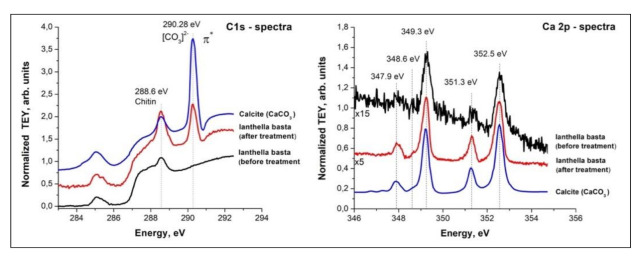
NEXAFS C1s (left) and Ca2p (right) spectra of *I. basta* skeletal fibers before and after NaOH treatment in comparison with calcite standard.

**Figure 10 ijms-22-12588-f010:**
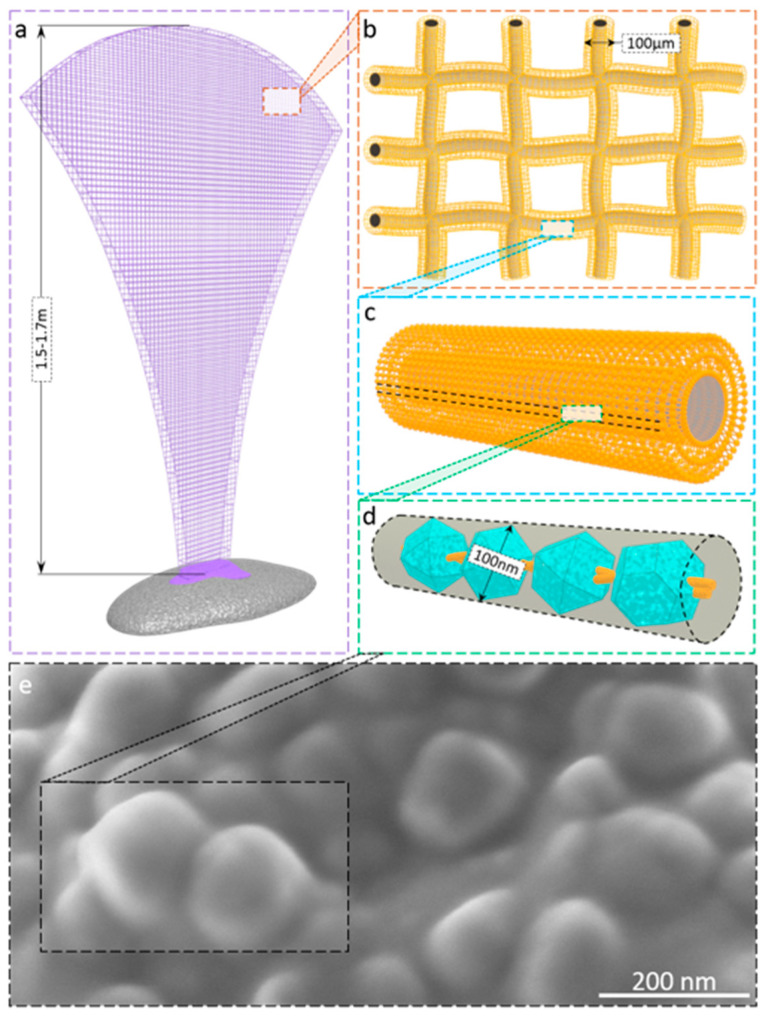
Schematic representation of mineralized giant network-like skeleton of *I. basta* marine demosponge (**a**). The flat construct is made of highly organized intercalated meshes of microfibers with a diameter of 100 µm (**b**). These fibers are made of nanostructured chitin (**c**) that plays the role of template for the formation of calcitic nanocrystals (**d**), which remains well visible using SEM (**e**).

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
