# Peer review of "Calcite Nanotuned Chitinous Skeletons of Giant Ianthella basta Marine Demosponge"

_ijms, 2021, doi:10.3390/ijms222212588_

Round 1

Reviewer 1 Report

This is an interesting manuscript. The authors mainly aimed to dissect the structure and composition of the demosponge skeleton. The reviewer has no major concern about the experimental design and data presentation but has a minor one. The reviewer noticed that the co-authors of this manuscript also published the similar identification work about skeleton in demosponge (Express Method for Isolation of Ready-to-Use 3DChitin Scaffolds from Aplysina archeri (Aplysineidae:Verongiida) Demosponge). However, this work was not cited. Compared to the previous co-author’s work, what is the major improvement and new findings for the present manuscript work? This should be discussed properly.

Reviewer 2 Report

A few notes are stated directly in MS

Nice work !!

The topic of the study is relevant and interesting for publication in International Journal of Molecular Sciences with some minor revisions. The study was conducted and written in a scientific and concise manner.
The authors have conducted studies to explore the nature and nanostructural organisation of the three-dimensional skeletal microfibers of the giant marine demosponge Ianthella basta. Although ianthellids have been intensively studied in many fields, there are still no reports of biominerals within the skeletal structures of ianthellids, including the demosponge I. basta.
The introduction is clearly stated, and the aim of the paper is briefly described at the end. M&M are thoroughly written and include detailed explanations of each analysis/subpoint. Results are clearly presented and interpreted. For the first time, using the battery of analytical tools mentioned in MS, the authors have shown that the biomineral calcite is responsible for nanostructuring the skeletal fibres of the sponge species studied. According to the authors, the origin of calcium, which is necessary for the formation of calcite, is still unclear, opening a wide field for further future research, in addition to investigating the natural selection of this species with respect to calcite.
The discussion appropriately contextualises the results of a study and focuses on the significance of the most notable findings.
The references are up-to-date and comprehensively written. The English language is correct to some degree and requires some light additional lecture by the author. Some suggestions are made in the text to rephrase the sentences.

Reviewer 3 Report

The rather substantial scientific work does not completely clarify the applicative purpose of the research. 
We ask you to make it clear and detailed both in the abstract and in the text.

Reviewer 4 Report

IJMS-1451646

In this manuscript, for the first time, the nature and nanostructural organization of three-dimensional skeletal microfibers of the giant marine demosponge Ianthella basta, with sponge specimens collected from two different geographical areals, were elucidated. This sponge has a micro-reticular body, consisting in a durable structure that determines the ideal filtration function of this organism.

For the first time, the biomineral calcite responsible for nano-tuning of the skeletal fibres of this sponge species was highlighted. This is the first report on the presence of calcitic mineral phase in representatives of verongiid sponges which belong to the class Demospongiae.

The manuscript is well written and the obtained experimental data suggest the possible role of structural aminopolysaccharide chitin as template for calcification.

The fact that biomineralization in the surface layers of chitin microfibers saturated with bromotyrosines have effective antimicrobial properties is of high interest to understand defence mechanisms of this organism and to investigate on antibacterial properties.

Revisions

Lines 2 and 3: “Ianthella Basta” change to “Ianthella basta”;

Line 91: “Hippospongia communis” change to Italic style;

Line 93: “I. basta” change to Italic style;

Line 116: “3D-visiulization” change to “3D-visualization”;

Line 147: “I. basta” change to Italic style;

Line 150: ‘… of Ca within them (Figure 4c,d).’ Figure 4 related to TEM comprehends Figure 4b and Figure 4c;

Line 151: “naocrystalline” change to “nanocrystalline”;

Line 156: “highlighted in greencolor” is it an autofluorescence?;

Line 161: Should Figure 6 be included in the manuscript after paragraph 3.2, after the citation of Figure 6 in the text?;

Line 279: “I. basta” change to Italic style;

Line 303: “Ianthella basta” change to Italic style;

Line 316: “I. basta” change to Italic style;

In the References section, change the names of the species in the titles of the references to Italics.
